# A Comprehensive Evaluation of Salt Tolerance in Tomato (Var. Ailsa Craig): Responses of Physiological and Transcriptional Changes in RBOH’s and ABA Biosynthesis and Signalling Genes

**DOI:** 10.3390/ijms23031603

**Published:** 2022-01-29

**Authors:** Abdul Raziq, Yu Wang, Atta Mohi Ud Din, Jin Sun, Sheng Shu, Shirong Guo

**Affiliations:** 1Key Laboratory of Southern Vegetable Crop Genetic Improvement, Ministry of Agriculture, College of Horticulture, Nanjing Agricultural University, Nanjing 210095, China; raziqbaloch867@gmail.com (A.R.); ywang@njau.edu.cn (Y.W.); attajutt82@yahoo.com (A.M.U.D.); jinsun@njau.edu.cn (J.S.); shusheng@njau.edu.cn (S.S.); 2Directorate of Vegetable Seed Production, Agriculture Research Institute, Village Aid Sariab, Quetta 87300, Pakistan; 3Suqian Academy of Protected Horticulture, Nanjing Agricultural University, Suqian 223800, China; 4Key Laboratory of Crop Physiology Ecology and Production Management, Ministry of Agriculture, College of Agriculture, Nanjing Agricultural University, Nanjing 210095, China

**Keywords:** tomato, ABA, RBOH, ROS, osmoprotectants, metabolites, antioxidant enzymes, reactive oxygen species, photosynthesis

## Abstract

Salinity is a ubiquitous stressor, depleting osmotic potential and affecting the tomato seedlings’ development and productivity. Considering this critical concern, we explored the salinity response in tomato seedlings by evaluating them under progressive salt stress duration (0, 3, 6, and 12 days). Intriguingly, besides the adverse effect of salt stress on tomato growth the findings exhibited a significant role of tomato antioxidative system, RBOH genes, ABA biosynthesis, and signaling transcription factor for establishing tolerance to salinity stress. For instance, the activities of enzymatic and non-enzymatic antioxidants continued to incline positively with the increased levels of reactive oxygen species (O_2_^•−^, H_2_O_2_), MDA, and cellular damage, suggesting the scavenging capacity of tomato seedlings against salt stress. Notably, the RBOH transcription factors activated the hydrogen peroxide-mediated signalling pathway that induced the detoxification mechanisms in tomato seedlings. Consequently, the increased gene expression of antioxidant enzymes and the corresponding ratio of non-enzymatic antioxidants AsA-GSH suggested the modulation of antioxidants to survive the salt-induced oxidative stress. In addition, the endogenous ABA level was enhanced under salinity stress, indicating higher ABA biosynthesis and signalling gene expression. Subsequently, the upregulated transcript abundance of ABA biosynthesis and signalling-related genes suggested the ABA-mediated capacity of tomato seedlings to regulate homeostasis under salt stress. The current findings have revealed fascinating responses of the tomato to survive the salt stress periods, in order to improve the abiotic stress tolerance in tomato.

## 1. Introduction

Salt stress is a major abiotic stress that adversely affects crop growth, development, quality, and yield [1,2]. High salinity affects plant cellular function by influencing uptake and assimilation of mineral ions, enzyme activity and photosynthetic function [3,4,5]. The higher accumulation of Na^+^ ions induces ionic and osmotic stress in plants, which causes alteration in plants structural and functional stability [6,7]. High salt concentrations in plants can cause an imbalance between production and scavenging of reactive oxygen species (ROS) such as superoxide anion (O_2_^•−^), hydrogen peroxide (H_2_O_2_) and hydroxyl radicals (OH^•^), particularly in chloroplasts and mitochondria, and induce hyperosmotic stress that can lead to oxidative damage [8,9]. In addition, the excess of salts reduces growth, mainly by reducing cell expansion in root tips and younger leaves thereby promoting premature senescence or programmed cell death [10,11].

Plants have developed various physiological, biochemical, and molecular processes to regulate salt stress’s adverse effects. Usually, plants endure sub-optimal environments by adopting a set of mechanisms that collectively respond against the stressful environment either by acclimation, escape, or detoxification [7,12]. Among them, one such possible mechanism is the accumulation of osmoprotectants, such as amino acids (proline) and non-structural sugars to maintain the osmotic balance under the prevailing stress conditions [13,14]. Besides this, plants also have an antioxidative defense system that contains enzymatic antioxidants (Superoxide dismutase (SOD), peroxidase (POD), catalase (CAT), ascorbate peroxidase (APX), Glutathione peroxidase (GPX), glutathione reductase (GR)) and non-enzymatic antioxidants (Ascorbate(AsA) and Glutathione (GSH)) that helps in scavenging the stress-induced reactive oxygen species (ROSs) [15,16].

ROS also acts as a multipurpose signal to initiate a series of subsequent defense signalling under various environmental stresses, including high salinity, drought, pathogenic infection and heat stress [17]. Particularly, the NADPH oxidase/respiratory burst oxidase homolog (RBOHs) transcription factors are essential in regulating ROS generation and subsequent signalling events for salinity adaptation. Previous investigations revealed that RBOHs are involved in different signalling pathways that regulate root hair growth, stomatal closure, pollen–stigma interactions, defense responses to pathogens, and acclimation to abiotic stresses [18,19,20]. Notably, tomato consists of eight RBOHs that are involved in stomatal movements through different phytohormones. For instance, *RBOH1* gene is linked to Abscisic acid (ABA)-mediated salinity tolerance in tomatoes. Accumulation of ABA usually increases transcript levels of *RBOH1* and other defense-related genes resulting in elevated apoplastic H_2_O_2_ collection, increasing the activity of NADPH oxidase and antioxidant enzymes in tomatoes [21]. In addition to their importance in biotic and abiotic responses, RBOH genes are also shown to play critical roles in plant growth and development associated with hormone signalling [22].

ABA is a primary plant hormone involved in salt stress response regulation, including stomatal closure, ion homeostasis, salt stress-responsive gene expression, and metabolic changes [23]. ABA functions as a central integrator that links and reprograms the complex developmental process under salt stress and activates adaptive signalling cascades in plants [24]. Under stress conditions, plants induce the production of ABA biosynthesis genes, such as Nine-Cis-Epoxycarotenoid Dioxygenases (NCEDs) and ABA Deficient (ABAs). The ABA receptors then perceive ABA Pyrabactin Resistance/Pyrabactin Resistance Like (PYR/PYL), which induce phosphorylation activity of the ABA-dependent Sucrose Non-Fermenting Related Protein Kinases (SnRKs) family, and the activation of ABA-dependent transcriptional network involved in ionic and osmotic adjustments in response to salt stress [25,26,27].

Tomato (*Solanum Lycopersicum*), a widely used vegetable crop throughout the world, is a rich source of antioxidant molecules such as carotenoids, vitamins E and C, ascorbic acid and phenolic compounds, mainly flavonoids. However, salt stress imposes several negative effects on the germination, growth, biomass accumulation and yield of tomatoes [11]. Keeping in view the increasing salinity concerns and global importance of tomato crop, this study was designed with an objective to demonstrate the role of the tomato antioxidative system, RBOH genes, ABA biosynthesis and signaling transcription factor under 12 days long salt stress to provide the basis for future studies regarding the improved salinity tolerance of tomato crops.

## 2. Results

### 2.1. Salt Stress Affects Morphological Indices and Pigment Contents of Tomato Seedlings

Salt stress significantly affected the growth rate of tomatoes as depicted by the linear decline in pigment contents and the seedling vigour. Compared to control, the contents of Chl-a, Chl-b, Total Chlorophyll and Carotenoids contents decreased by 57.11%, 41.47%, 51.14% and 15.75%, respectively (Figure 1B–E). Consistently, the tomato seedlings’ morphological indices, including biomass accumulation stem diameter and plant height, were also severely affected by the increasing salt stress duration as depicted by the significantly declining seedling index (SI) value and root shoot ratio. The SI value of seedlings decreased by 44.41%, 63.96%, 76.53% after 3, 6 and 12 days of salt stress, respectively (Figure 1A).

### 2.2. Salt Stress Alters the Plant-Water Relations and Osmotic Potential of Tomato Seedlings

The impact of salinity on the plant-water relationship was calculated by comparing the water potential (Ψw), osmotic potential (Ψs), turgor potential (Ψp) and osmotic adjustments (OA) between the control and salt treatments. As the salt stress duration increased, the values of Ψw and Ψs became more negative. Compared to control, the value of Ψw and Ψs significantly reached up to −111.24% and −290% after 12 days of salt stress (Figure 2). Simultaneously, compared to control, the value of Ψp increased up to 74.26% after the 12 days of salt stress (Figure 2A–C). Consistent with the alterations in Ψw, Ψs and Ψp, salt stress-induced significant (92%) osmotic adjustments in tomato seedlings. Furthermore, the relative water content was decreased up to 63.33% under salt stress duration (Figure 2E).

### 2.3. Salt Stress Changes the Protein, Soluble Sugars, and Proline Content in Tomato Seedlings

To assess the osmoprotective potential of tomato plants after salt stress, we measured the levels of two significant osmolytes, soluble sugars and proteins, in the roots and leaves of tomato seedlings. Salt stress induced the significant accumulation of proline and soluble sugars in the leaves and roots of tomato seedlings. Compared to control, the quantity of soluble sugars and proline increased by 93.07% and 79.77% in roots, and 56.02% and 71.00% in leaves, after 12 days of salt stress, respectively (Figure 3B,C,E,F). However, the salt stress negatively affected the protein contents, and a significant decline (65.09% and 92.43%) was observed in root and leaves after 12 days of salt stress, respectively (Figure 3A,D).

### 2.4. Salt Stress Disturbed the Ionic Ratios and the Contents of Macro and Micronutrients in Tomato Seedlings

Salt stress significantly changed the nutrient contents of macro and micronutrients in both leaves and roots. Compared with control, the contents of phosphorus (P), iron (Fe), manganese (Mn), magnesium (Mg), copper (Cu) and potassium (K^+^) decreased significantly while the contents of zinc (Zn) and calcium (Ca) increased significantly under the salt stress (Table 1 and Table 2). P content decreased by 30.43% and 51.41%, Fe contents decreased by 30.36% and 54.27%, Mn contents decreased by 83.33% and 40.28%, Mg contents decreased by 35.97% and 39.40% and K^+^ decreased by 27.35% and 28.87% in leaves and roots after 12 days of salt stress, respectively. However, the Cu, Ca^2+^ and Na^+^ contents increased by 119.77% and 35.84%, 124.04% and 71.82%, and 36.86% and 87.85% in leaves and roots after the 12 days of salt stress, respectively. Overall, the impact of salt stress was most obvious in the roots of salt-treated tomato seedlings than the leaves.

### 2.5. Salt Stress Instigates ROS Accumulation, Lipid Peroxidation and Oxidative Damage in Tomato Seedlings

The oxidative stress induced by the progressive salinity was estimated by calculating the accumulation of H_2_O_2_ and O_2_^•−^ in the leaves and roots of tomato seedlings. A significant linear increase was observed in the amounts of H_2_O_2_ and O_2_^•−^ with increasing duration of salt stress in both leaves and roots. H_2_O_2_ and O_2_^•−^ levels increased by 46.31% and 96.40% in roots, and 57.18% and 87.84% in leaves, after 12 days of salt stress, respectively (Figure 4A,B,E,F). Furthermore, we confirmed the oxidative damage induced by salt stress in roots by root activity analysis (TTC staining) and leaves by MDA content and membrane stability index (MSI). Consistent with the ROS accumulation, the MDA contents increased significantly with increasing salt stress compared to control.

Similarly, the increasing lipid peroxidation and oxidative stress seriously damaged the membrane stability, as depicted by the 67.19% decline in MSI, after 12 days of salt stress (Figure 4C,D). Also, the root activity markedly declined by 57.23% compared to the control after 12 days of salt stress (Figure 4G). We further checked the gas exchange parameters and the photochemical efficiency, by carefully expanding the wilted leaf with hand, to assess the photosynthetic inhibition under salt-induced oxidative stress and found a linear decrease with increased salt stress duration (Appendix A).

### 2.6. Salt Stress Activates the ROS Scavenging Mechanism in Tomato Seedlings

The capacity of tomato seedlings to counter the increasing accumulation of ROS to protect the cells from further oxidative damage under salt stress was evaluated by measuring the activities of major enzymatic and non-enzymatic antioxidants. Generally, salt stress triggered the activities of all the studied antioxidant enzymes to counter the production of excessive ROS. However, after an initial increase (128.47% and 405.55%) up to 6 days compared to control, the activity of SOD and CAT decreased significantly after 12 days of salt stress. Compared to control, the plants after 12 days of salt stress had 472.72%, 321.12%, 62.00%, 166.68%, 62.03% and 176.67% more activities of POD, APX, GR, MDHAR, DHAR and MDAR, respectively (Figure 5A–H). We further verified the response of antioxidant enzymes by checking the relative expression levels of genes encoding antioxidant enzymes. The gene expression level consistently showed a similar trend and confirmed the salt-stress induced upregulation in the gene expression level of antioxidant enzymes. The genes expression levels of leaves and roots revealed the upregulation in activities of SOD (1.73, 2.78-fold), POD (5.69, 2.27-fold), CAT (2.12, 2.50-fold), APX (2.62, 4.01-fold), GR (4.80, 2.31-fold), MDHAR (4.09, 4.65-fold), and DHAR (2.99, 7.75-fold) after 12 days of salt stress, respectively as compared to control (Figure 6A,B).

Similar to enzymatic antioxidants, salt stress also induced alterations in the contents of non-enzymatic antioxidants, namely AsA, DHA, GSH and GSSG, and their ratios. The contents of AsA increased significantly by 2.13, 179.48 and 554.13%, while DHA content decreased by 16.05, 17.99 and 151.25% after 3, 6 and 12 days of salt stress (Figure 6A,B). Consequently, the ratio of AsA/DHA increased markedly with the increasing duration of salt stress. Similarly, the contents of GSH, GSSH and their ratios also showed a significantly increasing trend. GSH content increased by 10, 25 and 45%, while GSH content increased by 81.81%, 113.33 and 175.86%, compared to control after 3, 6 and 12 days of salt stress, respectively (Figure 7A,B).

### 2.7. Salt Stress Enhanced the Activation of Polyamines Metabolism Enzymes in Tomato Seedlings

Under salt stress, the activities of three major polyamine metabolism-related enzymes, ADC, DAO, and PAO, were measured in tomato leaves. The findings revealed that increasing the duration of salt stress significantly increased PA metabolism enzymes’ activity. The activities of two biosynthesizing enzymes, ADC and DAO, and catabolizing enzymes (PAO), increased by 243.42%, 205.00%, and 222.00%, respectively, when compared to the control (Figure 8A–C).

### 2.8. Salt Stress Upregulated Expression of Antioxidant, RBOHs and ABA Biosynthesis and Signalling Related Genes in Tomato Seedlings

The transcripts abundance of various RBOHs related transcription factors was analyzed through qRT-PCR analysis. The results showed that 12 days of salt stress caused significant up-regulation of *SlRBOH1*, *SlRBOH-A*, *SlRBOH-D*, *SlRBOH-E*, *and SlRBOH-F* in the leaves, and *SlRBOH1*, *SlRBOH-A* and *SlRBOH-D* in the roots of tomato seedlings, as compared to the respective control plants. Interestingly the transcription level of *RBOH-H* in leaves, *RBOH-E*, *RBOH-F* and *RBOH-H* in roots did not show significant variations in their expression levels during the early days (three days in leaves, six days in roots) of salt stress (Figure 9A,B).

In tomato seedlings under salt stress, the ABA content increased with the increasing duration of the salt stress and the highest ABA contents in both leaves and roots was noticed after 12 days of salt stress treatment (Figure 10A,B).

Therefore, we further analyzed the expression of ABA biosynthesis and signaling genes. Under salt stress, the transcription levels of ABA biosynthesis and signalling genes showed higher expression levels than control. For instance, salt stress upregulated the expression of ABA biosynthesis genes including *SlZEP*, *SlNCED1*, *SlNCED3*, *SlNCED5*, *SlAAO3*, *SlABI3* and *SlABI5* in both leaves and roots, as compared to the control (Figure 11A,B).

Similarly, the ABA signaling genes, *SlSnRK2.2*, *SlSnRK2.3*, *SlSnRK2.6*, *SlPYL4*, *SlPYL8*, *SlABF4* and *SlDREB2* also showed significantly higher transcript levels in leaves and roots in response to salt stress. Taken together, it suggested that the plants tried to counter the damage via enhanced signalling and biosynthesis of ABA (Figure 12A,B).

## 3. Discussion

Soil salinity is an increasingly severe global problem, as salt hampers plant growth and development and reduces crop yield. In addition to naturally occurring soil salinity, salinization increases due to irrigation practices and climate change [27]. Therefore, in the present study, we explored the response of the tomato seedlings to three different durations of salt stress. Despite the salt-induced oxidative damage, we found that tomato seedlings adopted several strategies, including antioxidative scavenging of ROS and activation of stress-responsive gene signalling to detoxify oxidative stress.

One of the most common symptoms of plants under a stressful environment, e.g., salt or drought stress, includes leaf yellowing and growth inhibition, as plants serve their energy on survival rather than growth improvement [28]. Similarly, in the present study, increasing salt stress duration induced significant loss of photosynthetic pigments and markedly affected the morphological indices of the tomato seedlings (Figure 1). Such losses in pigment appear to be caused by the reduced activity of chlorophyll biosynthesis enzymes and increased chlorophyllase activity that degraded the chlorophyll content under salt stress [29,30]. In addition, sodium ions could disrupt the uptake of some important cation, i.e., Mg^+2^ that is the main part of the chlorophyll molecule, thus impacting the chlorophyll content of the leaves. Consequently, it impaired the growth and biomass accumulation in the roots and leaves of tomato seedlings. Consistent with our results, number of previous studies found the loss of pigment and growth inhibition of plants under salt stress [7,31,32].

The excess accumulation of sodium ions under the salt stress imposes osmotic stress, disrupting the water permeability and causing a disturbance in the water potential and turgidity of the cells [33]. Consistently, in the present study we also noticed that the values of Ψw and Ψs become more negative, enhancing the corresponding Ψp and osmotic adjustments. Ultimately, the RWC declined significantly (Figure 2). Such reduction in RWC of the cells are the primary indicator of the water stress that limits the water flow to the new cell elongation sites [13]. Similar results were reported earlier in peach [34] and tomato [35].

Moreover, salt stress-induced reduction in the water status of the cell results in stomatal closure, inhibiting the gaseous exchange and photosynthetic ability of the leaves [36]. Combined with the decreased pigment content, it ultimately impairs the photochemical efficiency of the plants. We consistently observed decreased values of the gas exchange parameter and maximal photochemical efficiency (Fv/Fm) in tomato seedlings (Appendix A), indicating the severe consequences of salt-induced osmotic stress on tomato seedling photosynthetic efficiency [36,37]. However, the marked increase in osmotic adjustments in tomato seedlings under salt stress also elucidates that it strives to counter the salinity-imposed osmotic stress by modifying the plant-water relations.

Osmotic regulation is a crucial self-defence mechanism produced by plants under stress. Plants produce high levels of osmoprotectants (e.g., soluble sugars, soluble proteins and proline content) to maintain the osmotic pressure and reduce cell water loss [38]. Proline and soluble sugars accumulated in response to stressful conditions act as a low molecular weight antioxidant that detoxifies the stress-induced toxicity and contributes to cellular osmotic adjustments [39]. In previous studies with various abiotic stresses, including salinity, these osmolytes increased significantly in chickpea [40], *Amaranthus tricolor* varieties [41], grapes [13] and tomatoes [42]. In addition, burst production of ROS under salt stress could destabilize the protein metabolism, decreasing the contents of soluble proteins in the plants [43]. Consistent with these reports, in our study, protein content decreased. At the same time, proline and soluble sugars were increased with increasing salt stress duration, suggesting that tomato plants suffered salt-induced toxicity and regulated their osmolytes to survive the water limitations under salt stress (Figure 3).

The declined growth parameters and disturbed plant-water relations due to salinity stress indicate a need to study the plant cells’ sodium accumulation and minerals contents.

In the present study, the salt stress obviously enhanced the Na^+^ accumulation in the roots and leaves of tomato seedlings, disturbing the Na^+^/K^+^ ratio and ion homeostasis [44,45]. It could be due to the activation of the guard cells’ outward rectifying K-channels (GORK), which enhanced the Na^+^ ion accumulation by the outflow of K^+^ and resulted in declined K^+^ levels [44]. Besides the osmotic stress, the higher sodium accumulation in cells could also alter the uptake of other minerals, causing an imbalance in the contents of macro and micronutrients of the plants [45,46,47,48]. Similarly, in the present study we noticed a significant decline in the contents of important minerals like P, Mg, Mn and K, which is consistent with the decreased growth and chlorophyll contents of the tomato seedlings under salt stress.

In addition, the significant increase in the contents of heavy metals like Cu and Zn under the salt stress could be attributed to the loss of ion specificity due to excessive salt accumulation in the roots and leaves of tomato seedlings [49]. Previous studies have reported the increase in Ca^2+^ ions as a response to salt stress to regulate plant response against salt stress [50]. In addition, the wheat genotypes with high salinity tolerance maintained the higher calcium content against salt stress, suggesting the important role of Ca ions in maintaining the ion homeostasis and enzyme activities under salt stress [49]. Consistently, our results also showed higher Ca content in tomato seedlings which could be a response to enduring the salt stress. Collectively, these results showed that salt stress negatively affected the macro and micronutrients in the tomato seedlings. Similar trends were noticed in previous studies with different plant species [51,52,53,54].

ROS production, MDA accumulation and electrolyte leakage are the basic biomarkers for oxidative stress under abiotic stresses [55,56]. In the present study, salt stress induced the production of excessive O_2_^•−^ and H_2_O_2_. Consequently, the higher oxidative stress caused lipid peroxidation and MDA accumulation leading to cell membrane damage and increased leakage of electrolytes as compared to control plants. These results are similar to those in tomato [36], grape [13], and chickpea [40] and demonstrate the severe consequences of salt stress on the growth and development of plants (Figure 4).

Plants have developed different tools against stress which could regulate redox homeostasis and protect plant cells from oxidative damage by scavenging excessive ROS [57]. For instance, SOD is one of the most important antioxidant enzymes that carry out the dismutation of superoxide anion (O_2_^•−^) into oxygen (O_2_) and comparatively less toxic ROS, i.e., hydrogen peroxide (H_2_O_2_). Subsequently, the other major antioxidant enzymes like POD, APX and CAT detoxify the H_2_O_2_ to water [58,59,60]. In the present study, salt stress-induced activities of various enzymatic (SOD, POD, CAT, APX, GR) and non-enzymatic antioxidants (AsA, GSH) in the tomato seedlings. These results are in line with the previous reports in tomato [30] and other species like *Solanum Lycopersicum* [36], *O. sativa* [61], *C. arietinum* [30], *B. juncea* [62,63] and wheat [64]. It suggested that tomato seedlings retained the capacity to tolerate or detoxify the salt-induced toxicity for their survival. Another important antioxidants-mediated protective mechanism in plants includes AsA-GSH cycle that assists the antioxidant enzymes to scavenge the excessive ROS. Both AsA and GSH are potent antioxidants and are considered the buffering agents in redox reactions to protect the plasma membrane from stress-induced oxidation [65]. In this perspective, the H_2_O_2_ produced after superoxide dismutation is detoxified by APX into H_2_O using AsA as the substrate. In the present study, the increased GR activity after salt stress provides GSH, reducing DHAR to dehydroascorbate (DHA) and then to AsA via the AsA–GSH cycle. Similarly, GSH is oxidized to GSSG and subsequently recycled by GR. Thus, the ratio of GSH/GSSG is essential for sustaining the cell redox state [66]. The increase in the activities of SOD, POD, CAT, APX, GR, MDHAR, DHAR, and AsA, DHA was observed in the current study. The ratio of AsA/DHA, GSH, GSSG, and the ratio of GSH/GSSG and proline content in tomato seedlings was observed to be affected by salinity (Figure 5, Figure 6 and Figure 7), implying that tomato seedlings attempted to maintain redox regulation under salt stress. It was also confirmed at the transcriptomic level, as the relative transcript levels of antioxidant enzymes showed a similar trend with increasing salt stress duration. A similar gene expression pattern was previously observed in tomato under high temperature stress [67,68].

In addition, salinity stress also caused a significant increase in polyamines metabolism as suggested by the increase in the activities of three key enzymes named DAO, PAO and ADC (Figure 8). Similarly, in previous reports, salinity stress enhanced the ADC and ODC activities in tomato roots, indicating that both enzymes are responsible for stress tolerance [69]. Furthermore, salinity stress also induced a slight increase in PAO activity [70]. It has been well known that polyamines play important role under stress conditions by modulating ROS homeostasis and regulating the antioxidative mechanism to suppress the ROS production under stress conditions [71]. Therefore, in the present study, the significant increase in enzymes involved in polyamines synthesis could be linked with the signaling regulation to enhance the responsiveness of tomato seedlings against salt stress (Figure 8).

The plasma membrane-localized respiratory burst oxidase homolog (RBOH) proteins generate reactive oxygen species (ROS). *AtrbohD* and *AtrbohF*, the major NADPH oxidases responsible for ABA-induced ROS production in Arabidopsis, are involved in stomatal closure [72]. *OST1* interacts with *AtrbohD* and *AtrbohF* and phosphorylates Ser174 in *AtrbohF* [73]. To acquire further molecular insights into the salt tolerance mechanism of WT Ailsa Craig, we tested the expression of several TFs, ABA biosynthesis enzymes and signalling and defence-related proteins in WT seedlings. We noticed a significant upregulation of all the studied RBOH genes with the increasing salt stress duration (Figure 9). Combined with the accumulation of hydrogen superoxide radicals, it could be suggested that salt-induced over-production of H_2_O_2_ activated the RBOH signalling mechanism, which regulated the various adaptive mechanisms to counter the salt-induced osmotic stress in tomato seedlings. In previous studies with different abiotic stresses, including low temperature [74], cold [74] and salinity [75], the acclimation to the stresses was found associated with the increased transcript levels of *RBOH1* [76]. Furthermore, in Arabidopsis, the expressions of *AtrbohD* and *AtrbohF* were upregulated and Na^+^/K^+^ homeostasis in wild-type Arabidopsis seedlings that grow on MS medium containing NaCl [77]. Previous studies with sugar beet revealed that *BvRBOHE*, *BvRBOHF*, and *BvRBOHH* downregulates under salt stress. However, we found significant upregulation of both *SIRBOH-E*, *SIRBOH-F* after 12 days of salt stress and only *SIRBOH-H* was found downregulated in roots after 12 days of salt stress (Figure 9). It suggests the significant role of RBOH transcription factors in tomato salt-stress acclimatization. Consistent with our results, a genome-wide identification of RBOH genes showed significant upregulation *RBOHA*, *RBOHD*, *RBOHF* and *RBOHG* under drought stress. Altogether, the significant upregulation of RBOH genes, particularly after 12 days of salt stress, strongly suggests their role in regulating the response of tomato seedlings to long-term salinity stress.

The phytohormone ABA plays a crucial role in regulating a range of plant physiological processes in response to various stresses [78]. ABA functions as an essential secondary signalling molecule to activate a kinase cascade and mediate gene expression during salt stress [79]. Osmotic stress, such as drought and high salinity, dramatically increases the ABA level, which induces the expression of many genes involved in stress responses [80]. In the present study, we consistently observed the ABA accumulation with the increasing duration of the salt stress (Figure 10), suggesting that tomato seedlings regulated the ABA levels and the expression of relevant genes for survival against the salt-induced damages. Previously, biochemical and genetic studies have shown that 9-cis–epoxycarotenoid dioxygenase (NCED) is a key rate-limiting enzyme in ABA biosynthesis and its overexpression in tomato and other plants causes abscisic acid (ABA) accumulation, affecting the stress responsiveness of plants [81,82,83]. Similarly, zeaxanthin epoxidase (*ZEP*) and aldehyde oxidase (*AAO3*) are also critical regulatory genes in the ABA biosynthesis pathway in Arabidopsis and other plant species [84]. Therefore, significant accumulation of ABA in roots and leaves of tomato seedlings suggests that salt stress induced the expression of ABA biosynthesis genes (*ZEP*, *ABI3*, *ABI5*, *NCED1* and *NCED2*) that upregulated the corresponding ABA abundance levels to regulate the plant growth and homeostasis under salt stress. Previously, *AtZEP*-overexpressing plants exhibiting vigorous growth, enhanced de novo ABA biosynthesis, increased the expression level of salt stress-related genes, and suggested the critical role of ABA biosynthesis and ABA signalling against salt stress [85]. Altogether, it shows that enhanced biosynthesis of ABA is a key regulatory strategy of tomato seedlings to survive under salt stress (Figure 11).

In addition to ABA-biosynthesis induction, salt stress also activated the downstream ABA signalling mechanism that controls ABA-regulated gene expression to enhance stress tolerance [86]. Various gene families like SnRks [87], DREBs [88,89], PYLs [90] and ABFs [91] are involved in ABA signalling that improves the tolerance of plants under stressful environments [92]. Among SnRKs, Subfamily 2 of SNF1-related protein kinase (*SnRK2*) is considered as a positive global regulator of abscisic acid signalling [87]. Previously, it was noticed that *SnRK2.2*, *SnRK2.6* and *SnRK2.3* activated the *AREB1*/*ABF2*, *AREB2*/*ABF4*, and *ABF3* as a response to osmotic stress at the vegetative stage [93]. Therefore, the significant upregulation of the *ABF4*, *PYL4* and *PYL8* in the present study could also be linked to the higher expression of SnrK genes to improve the ABA-regulated signalling against the salt stress. Consistently, a recent study showed the alterations in transcriptional levels of several salt stress-responsive genes like *SlPP2C37*, *SlPYL4*, *SlPYL8*, *SlNAC022*, *SlNAC042*, and *SlSnRK2* family by *PpSnRK1α*, signifying that SnRK1α is involved in the ABA signalling pathway to improve tomato salt tolerance [94]. Functional analysis of the AREB/ABFs revealed that these proteins were positive regulators of the ABA-dependent signalling pathways under drought conditions [95]. Similarly, physiological, biochemical and transcriptomic analyses showed that *SlDREB2* enhanced plant tolerance to salinity by improvement of K^+^/Na^+^ ratio and proline and polyamines biosynthesis [96]. Altogether, these reports suggested that the upregulation of ABA-signalling related genes in tomato seedlings regulated various downstream mechanisms involving the accumulation of osmolytes, osmotic adjustments and maintaining a turgor potential that could help the plant to survive under the 12 days long salt tress (Figure 12). Our results are consistent with the previous reports regarding the ABA-accumulation and signalling under the osmotic or salt stress [97,98].

## 4. Materials and Methods

### 4.1. Plant Material and Salt Stress Application

The tomato (*Lycopersicon esculentum* Mill.) the salt-sensitive cultivar, “Ailsa Craig” [99] was used in the present study. Seeds were collected from the Laboratory of Protected Horticulture, College of Horticulture, Nanjing Agricultural University, Nanjing, China and surfaced sterilized with 10% Sodium hypochlorite solution (NaClO), rinsed five times with deionized dd H_2_O (double distilled water) and incubated at 28 °C in Petri dishes under dark conditions until the emergence of the radicle. The seedlings were grown in half-strength Hoagland’s solution under controlled conditions (25 °C/18 °C, day/night temperature, 16-h light/8-h dark cycle and 40–50% relative humidity). Subsequently, at the four-leaf stage, tomato seedlings were subjected to high salinity stress treatment (120 mM NaCl solution) for 3, 6 and 12 days [100] while the control plants were grown under half-strength Hoagland’s solution without salt stress during this duration. Each treatment contained three biological repeats. At each sampling point, samples were collected in liquid nitrogen and stored at −80 °C for further physiological, biochemical and molecular analysis.

### 4.2. Determination of Seedling Index and Pigment Contents

Different morphological parameters like plant height and fresh and dry weight of plant parts (root, shoot, and leaves) were recorded using meter scale and electronic balance, respectively. After recording the fresh weight, the samples were dried in a hot air oven at 80 °C for 72 h to measure the dry weight. Subsequently, the seedling index (SI) was calculated as follows [75,101,102]:$$\mathrm{SI}=(\;\frac{\mathrm{Stem}\;\mathrm{diameter}}{\mathrm{Plant}\;\mathrm{height}}+\frac{\mathrm{Root}\;\mathrm{dry}\;\mathrm{weight}}{\mathrm{Shoot}\;\mathrm{dry}\;\mathrm{weight}}\;)\times \mathrm{tomato}\;\mathrm{seedling}\;\mathrm{dry}\;\mathrm{weight}$$

The chlorophylls and carotenoid pigments were extracted by grinding 0.5 g of fresh leaf sample with 80% acetone. Subsequently, the absorbance of the extract at 663.3 nm, 646.6 nm, 510 nm and 470 nm was analyzed by a UV-1800 spectrophotometer and pigment contents was calculated according to [103,104]

### 4.3. Measurement of Plant-Water Relations

The plant-water relations were measured by calculating the relative water content (RWC), leaf water potential (Ψt), osmotic potential (Ψπ), Turgor potential and Osmotic adjustment of the youngest completely expanded leaf. Measurements were performed at 3-h intervals from 9:00 a.m. to 6:00 p.m. Leaf water potentials (Ψt) were measured using a dew-point psychrometer (WP4, Decagon Devices, Washington) three times at 8:00 a.m., 1:00 p.m. and 5:00 p.m. The osmotic potential (Ψπ) was measured on frozen/thawed leaf samples and the pressure potential (Ψp) was estimated as the difference between Ψt and Ψπ, assuming a matric potential equal to 0. Leaf osmotic adjustment (OA) was determined as the difference Ψ π_0_V0 − Ψπ V, where Ψ π_0_V0 is the product of [osmotic potential] × [osmotic volume] of unstressed plants and ΨπV is the product of [osmotic potential] × [osmotic volume] of leaves from salinized plants. For each measurement, the osmotic volume was approximated by the corresponding relative water content value (RWC) calculated as [102]:$$\mathrm{RWC}=(\;\frac{\mathrm{Fresh}\;\mathrm{weight}-\mathrm{Dry}\;\mathrm{weight}}{\mathrm{Saturated}\;\mathrm{weighr}-\mathrm{Dry}\;\mathrm{weight}}\;)\times 100$$

### 4.4. Determination of Protein, Sugars and Proline Content

For protein extraction, 0.3 g fresh weight of samples were homogenized in a pre-chilled mortar with 600 uL of protein extraction buffer (50 mM Tris-HCl (pH 8.0), 150 mM NaCl, 1 mM EDTA, 1% (*v*/*v*) NP-40, 1% (*w*/*v*) sodium deoxycholate, 0.1% sodium dodecyl sulfate (*w*/*v*) and 1 mM PMSF). The extracted protein was quantified according to the Bradford kit (FD2003 by FDbio Science Biotech Co., Ltd., Hangzhou, China) [105].

Soluble sugar content was determined by using a specified sugar assay kit (145-1-1) and BCA assay kit (BCAP-1-W) by Nanjing Jiancheng Bioengineering Institute, Nanjing, China and Suzhou Comin Biotechnology Co., Ltd., Anhui, China, respectively as per their instructions [106].

For proline content, the fresh leaves sample was homogenized in 3% sulfosalicylic acid and the reaction mixture containing the extract, ninhydrin, glacial acetic acid (1:1:1) was incubated at 90 °C for 1  h. Subsequently, the toluene was added when the solution got cool, and absorbance was measured at 520 nm using a BioMate spectrophotometer [107].

### 4.5. Determination of Nutrient Elements Content in Leaves and Roots

Dried roots and leaves of tomato seedlings were oven-dried at 80 °C and ground using a mortar and pestle; 0.1 g of powder was then digested with 5ml of nitric acid for 3 h, and then nutrient elements concentrations were analyzed using an atomic absorption spectrophotometer (Varian spectra AA 220, Varian, Palo Alto, CA, USA), according to the method described by [108]

### 4.6. Quantification of Reactive Oxygen Species (ROS), Oxidative Damage and Root Activity

The oxidative stress was estimated by quantifying the amount of O_2_^•−^ and H_2_O_2_ as described in the previously published protocols [109]. O_2_^•−^ was measured by mixing the supernatant of the processed leaf sample with phosphate buffer (50 mM; pH 7.8) and hydroxylamine hydrochloride (10 mM). The absorbance was checked at 530 after the incubation of reaction mixture incubation for about a half-hour at room temperature. For O_2_^•−^, the leaf sample processed in TCA (0.1%) and centrifuged (12,000× *g*; 15 min; 4 °C). The obtained supernatant was mixed with KI (1 M) and phosphate buffer (0.1 M; pH 7.8) and absorbance was noted at 390 nm after the dark incubation for one hour.

We quantified the malonaldehyde (MDA) content to check the lipid peroxidation using the method of [109] to estimate the oxidative damage caused by salinity stress. In brief, 0.3 g fresh leaf sample was homogenized in % TCA (Trichloroacetic acid), centrifuged at 10,000× *g* for 5 min, 1 mL supernatant was mixed with % TBA (Thiobarbitu-ric acid), 30 min boiling, and the absorption was measured. In addition, the cell membrane integrity was estimated by calculating the membrane stability index (MSI) as described by [37,110].

The root activity was estimated following the TTC method as described by [111], and root activity was expressed as the capacity of root deoxidization (mg g^−1^ h^−1^).

### 4.7. Determination of Photosynthetic Rate, Chlorophyll Fluorescence

After salt stress for 12 d, the photosynthetic rate was measured with the portable photosynthesis system (LI-6400; Li-COR, Lincoln, NE, USA), maintaining the CO_2_ concentration at 380 μmol mol^−1^ and photosynthetic photon flux density at 1000 μmol m^−2^ s^−1^.

Tomato plants were dark-adapted for 30 min to measure the Fv/Fm with the Portable fluorometer (PAM-2100, Walz, Effeltrich, Germany) as previously described [112].

### 4.8. Assay of Enzymatic and Non-Enzymatic Antioxidants Activity

For the assay of antioxidant enzymes, the 0.2 g fresh leaf tissues were homogenized in phosphate buffer (50 mM, pH 7.8) in a pre-chilled mortar, followed by a centrifugation (12,000× *g* at 4 °C) for 20 min. The samples were stored at −80 °C for further analysis. SOD activity was checked by using nitro blue tetrazolium (NBT) [113]. 50% inhibition of photoreduction of NBT by enzyme activity was considered as one unit of SOD activity. POD activity was measured by the oxidation of guaiacol in a reaction mixture containing phosphate buffer (0.2 M, pH 6.0), guaiacol (50 mM) and hydrogen peroxide (2%) and absorbance was checked at 470 nm. CAT was measured according to the previously described protocols by the reduction of H_2_O_2_ as the decrease in absorbance decreased at 240 nm [114,115]. APX activity was induced in a mixture of phosphate buffer (50 mM, pH 7.0), 0.1 mM EDTA, 0.1 mM H_2_O_2_ and 0.5 mM ascorbate by adding the fraction of enzyme extract [116]. The change in absorbance was recorded at 290 nm. GR activity was measured using the specific kit by Solarbio Life Science, Beijing, China, according to the manufacturer’s instruction. The MDHAR (Monodehydro ascorbate reductase) and DHAR (dehydro ascorbate reductase) activities were estimated according to the previously described protocols at 340 nm and 265 nm, respectively [116,117].

For non-enzymatic antioxidants, i.e., ascorbate (AsA-DHA) and Glutathione (GSH-GSSG), the samples were prepared by homogenizing 0.2 g of leaf sample in 6% pre-chilled HClO_4_. The supernatant obtained after centrifugation (12,000× *g*; 15 min; 4 °C) of homogenized sample was stored at −80 °C for further analysis. Later on, the contents of AsA-DHA, and GSH-GSSG were obtained by following the previously described protocols [118,119,120].

### 4.9. Determination of Activities of Ornithine Decarboxylase (ODC), Arginine Decarboxylase (ADC) and Polyamine Oxidase (PA)

A total of 500 mg fresh plant tissue was homogenized in potassium phosphate buffer (100 mM, pH 8.0) containing PMSF (0.1 mM), pyridoxal phosphate (1 mM PLP), dithiothreitol (5 mM, DTT), 1mM EDTA, 10mM ascorbic acid and 0.1% PVP. After centrifugation at 12,000× *g* for 40 min at 4 °C supernatants were dialyzed and used for assaying the enzyme activity. Activities of ADC or ODC were determined according to [5] in an assay mixture containing tris-buffer (100 mM, pH 7.5), EDTA, pyridoxal phosphate (50 mM), DTT and 300 μL enzyme extract. After incubation at 37 °C for 2 min, 200 μL of L-arginine (for ADC) or 200 μL of L-ornithine (for ODC) were added and mixtures were again incubated for 1 h at 37 °C. After that, 5% perchloric acid was added and centrifuged again at 3000× *g* for 10 min. To 500 μL supernatant were added 2mM NaOH and benzoyl chloride, and the mixture was thoroughly mixed and incubated at 37 °C for 30 min. NaCl (2 mL) and ether (3 mL) were added the mixture was centrifuged again at 1500× *g* for 5 min. After extraction with ether, evaporated ether phase was redissolved in methanol (60%) and read at 254 nm.

To determine PAO activity fresh leaf tissue was homogenized in 100 mM phosphate buffer (pH 6.5) and the homogenate was centrifuged for 20 min at 10,000× *g* at 4 °C. Reaction mixture contained 100 mM phosphate buffer (pH 6.5), 200 μL 4-aminoantipyrine/N, N-dimethylaniline solution, 100 μL horseradish peroxidase and 200 μL enzymes extract. Change in optical density was monitored at 254 nm after initiating reaction by adding 20 mM of each spermidine and spermine [121].

### 4.10. Determination of ABA Content

ABA was measured in seedlings after 48 h of exposure to salt-stress tomato plants. Plant tissue (0.2 g FW) was homogenized with 5 mL extraction buffer (80% acetone, 100 mg ^−l^ butylated hydroxytoluene, 0.5 g^−l^ citric acid) and centrifuged for 5 min at 12,000× *g*. The supernatant was collected, dried and resuspended in 0.5 mL of TBS buffer (6.05 g^−l^ Tris, 0.20 mg^−l^ MgCl_2_ and 8.8 g/l NaCl, pH 7.8). ABA was quantified using an indirect ELISA [122]. ABA-BSA conjugates were prepared according to [123] as described [124]. ABA levels are expressed in ng (g FW) ^−1^.

### 4.11. Quantitative Real-Time PCR (qRT-PCR)

According to the manufacturer’s instructions, total RNA was extracted from tomato leaves and roots using the RNAsimple Total RNA Kit (Tiangen, DP419). The HiScript II Q RT SuperMix for qPCR (+gDNA wiper) Kit was used to reverse-transcribe total RNA (1 g) to cDNA (Vazyme, R223-01). The ChamQ SYBR qPCR Master Mix (Vazyme, Q311-02) was used in the StepOnePlusTM Real-Time PCR Method to conduct the quantitative real-time PCR (qPCR) assays (Applied Biosystems, United States of America). The PCR conditions consisted of denaturation at 95 °C for 3 min, followed by 40 cycles of denaturation at 95 °C for 10 s, annealing at 58 °C for 10 s, and extension at 72 °C for 20 s. The tomato actin gene was used as an internal control. Gene-specific primers were designed according to cDNA sequences, as described in Appendix A. Relative gene expression was calculated as described by [125].

### 4.12. Statistical Analysis

The data presented in the study was analyzed by the SPSS 20.0 (SPSS Inc., Chicago, IL, USA). The data is presented in the form of mean ± standard deviation according to Tukey’s test. The graphs and heat maps have been created using the Origin Pro 2021. At least three independent repeats were performed for each measurement. All the software used in the data analysis were provided by Nanjing Agricultural University, Nanjing, Jiangsu, China.

## 5. Conclusions

Salinity is one of the important constraints that adversely affect the productivity of the tomato plants. Therefore, in the present study, we evaluated the comprehensive response of tomato towards progressive salt stress treatments at physiological and molecular levels. It was noticed that salt stress significantly reduced the water content, growth and development of tomato plants and inhibited the photosynthesis. In addition, the oxidative damage and ROS accumulation increased concurrently with increasing salt stress duration. However, with increasing salt stress duration, we noticed some interesting stress response of tomato plants involving the accumulation of osmoprotectants like proline and soluble sugars, and upregulation of major enzymatic and non-enzymatic antioxidants to maintain the turgor potential and detoxify the excessive ROS production under 12 days long salt stress.

At the same time, ROS production and water limitation under salt stress also activated the various molecular factors including RBOH TFs, and ABA biosynthesis and signaling genes, respectively. These genes are well-known molecular players under stressful environment that regulate the ion hemostasis, osmotic adjustments and osmolytes (soluble sugars and proline) to protect plants from stress-induced damage. Therefore, in the present study, the combination of these stress responses employed by tomato plants could help to survive the salt stress. These results might provide the basis for the development of salt-tolerant tomato varieties for better production under the salt stress conditions. In addition, a comparative physio-molecular study of salt stress responses at seedling and reproductive stage could be a good subject for future studies to understand the survival capacity of tomato plants at different growth stages.

## Figures and Tables

**Figure 1 ijms-23-01603-f001:**
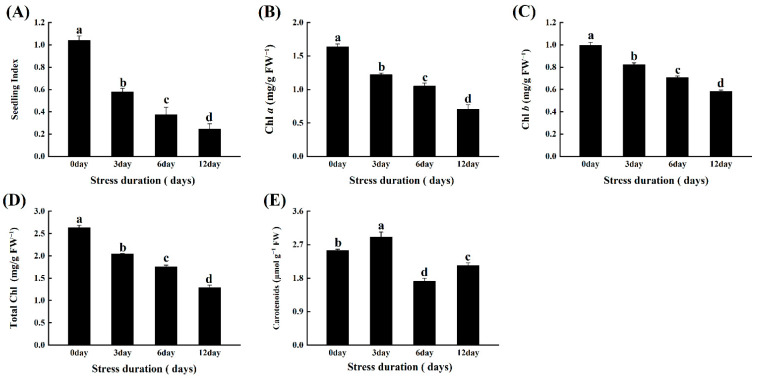
Effects of salt stress on (**A**) seedling index; (**B**) Chla; (**C**) Chlb; (**D**) Total Chl and (**E**) Carotenoids in leaves of tomato seedlings grown for 12 days under control and salt stress. Values are mean ± SD (Standard Deviation) of three replications. Different letters indicate significant difference at *p* ≤ 0.05 (Tukey’s HSD test).

**Figure 2 ijms-23-01603-f002:**
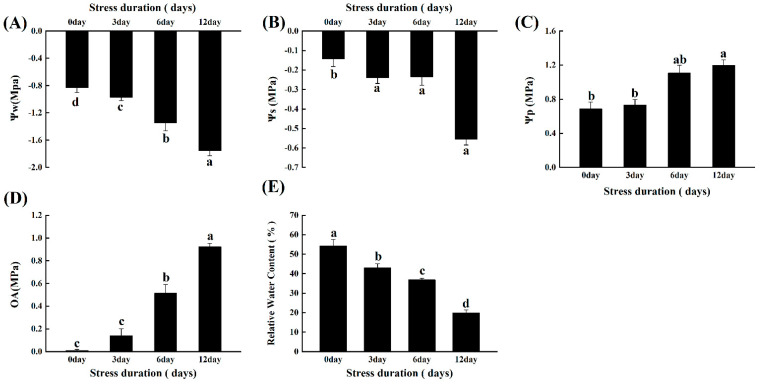
Effects of salt stress on (**A**) water potential (Ψw); (**B**) osmotic potential (Ψs); (**C**) turgor potential (Ψp); (**D**) osmotic adjustments (OA) and (**E**) relative water content (RWC) in leaves of tomato seedlings grown for 12 days under control and salt stress. Values are mean ± SD (Standard Deviation) of three replications. Different letters indicate significant differences at *p* ≤ 0.05 (Tukey’s HSD test).

**Figure 3 ijms-23-01603-f003:**
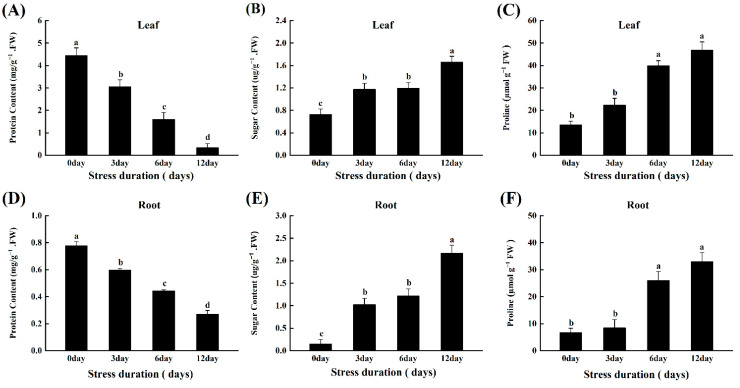
Effects of salt stress on protein content (**A**,**D**); Sugar content (**B**,**E**) and Proline (**C**,**F**) in leaves and roots of tomato seedlings grown for 12 days under control and salt stress. Values are mean ± SD (Standard Deviation) of three replications. Different letters indicate significant difference at *p* ≤ 0.05 (Tukey’s HSD test).

**Figure 4 ijms-23-01603-f004:**
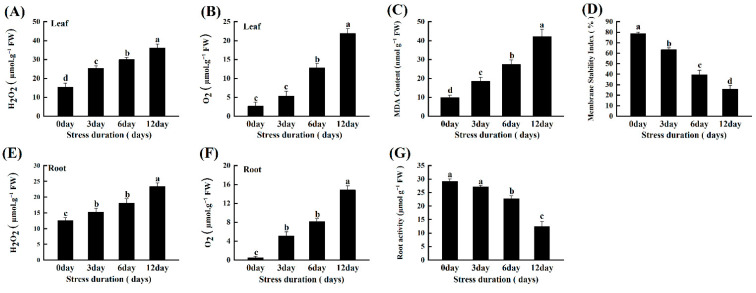
Effects of salt stress on H_2_O_2_ in leaves; roots (**A**,**E**); O_2_^•−^ in leaves; roots (**B**,**F**); malondialdehyde (MDA) (**C**); membrane stability index MSI (**D**); and root activity (**G**) of tomato seedlings grown for 12 days under control and salt stress. Values are mean ± SD (Standard Deviation) of three replications. Different letters indicate significant differences at *p* ≤ 0.05 (Tukey’s HSD test).

**Figure 5 ijms-23-01603-f005:**
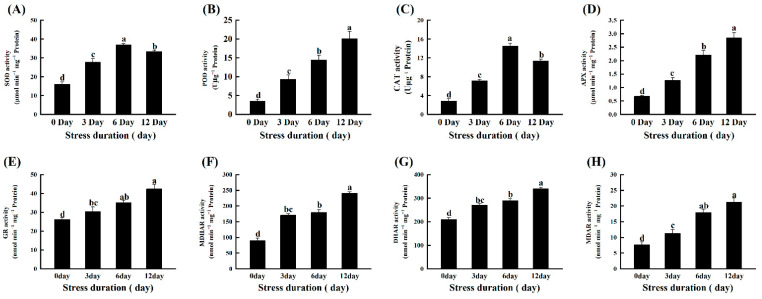
Effects of salt stress on antioxidant enzymes (**A**) superoxide dismutase SOD; (**B**) peroxidase POD; (**C**) catalase CAT; (**D**) ascorbate peroxidase APX; (**E**) glutathione reductase GR; (**F**) monodehydroascorbate reductase MDHAR; (**G**) dehydroascorbate reductase DHAR; (**H**) MDAR; in tomato leaves seedlings grown for 12 days under control and salt stress. Values are mean ± SD (Standard Deviation) of three replications. Different letters indicate significant differences at *p* ≤ 0.05 (Tukey’s HSD test).

**Figure 6 ijms-23-01603-f006:**
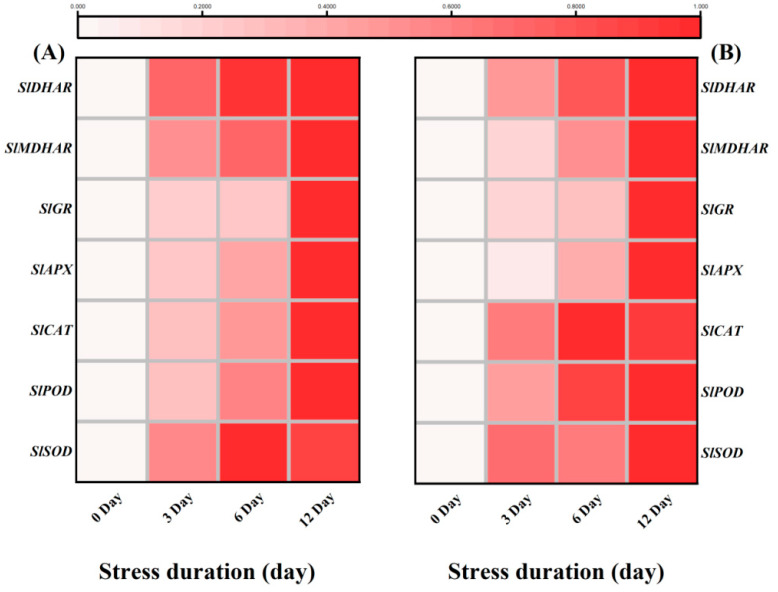
Heatmap of antioxidant enzymes related genes in tomato (**A**) leaves and roots (**B**) after 12 day of salt treatment. The scale (Log^2^ of the mean values after normalization; *n* = 3) shows the increase in relative concentrations from white to red color, compared to control (0 h).

**Figure 7 ijms-23-01603-f007:**
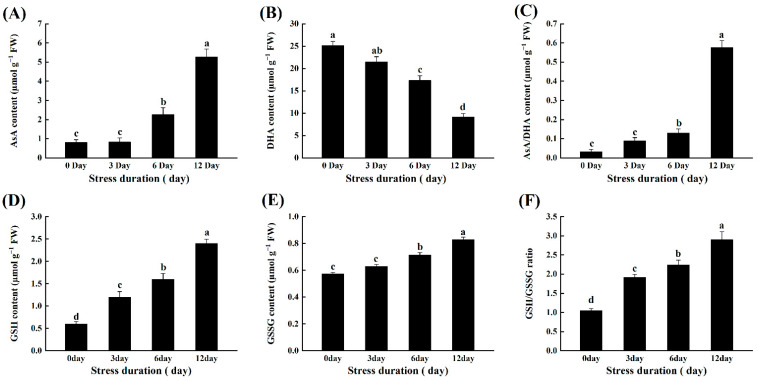
Effects of salt stress on non- antioxidant enzymes (**A**) Ascorbic acid AsA; (**B**) DHA; (**C**) AsA/DHA; (**D**) glutathione GSH; (**E**) GSSG; and (**F**) GSH/GSSG in tomato leaves seedlings grown for 12 days under control and salt stress. Values are mean ± SD (Standard Deviation) of three replications. Different letters indicate significant differences at *p* ≤ 0.05 (Tukey’s HSD test).

**Figure 8 ijms-23-01603-f008:**
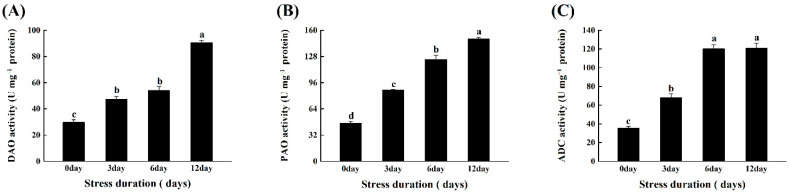
Effects of salt stress on (**A**) Diamine oxidase (ADC); (**B**) polyamine oxidase (PAO); and (**C**) Arginine decarboxylase (ADC) in tomato leaves grown for 12 days under control and salt stress. Values are mean ± SD (Standard Deviation) of three replications. Different letters indicate significant differences at *p* ≤ 0.05 (Tukey’s HSD test).

**Figure 9 ijms-23-01603-f009:**
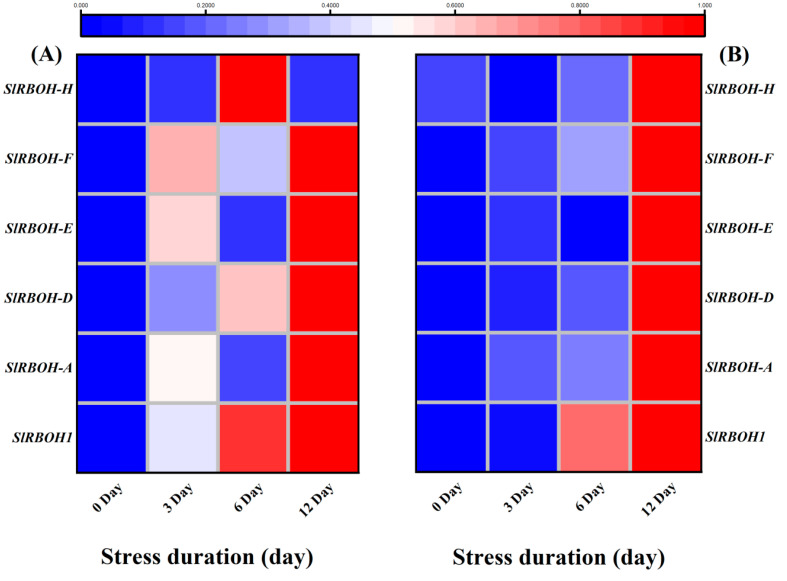
Heatmap of selected genes-related RBOHs (respiratory burst oxidase homolog proteins). The relative expression level was quantified from tomato (**A**) Leaves; (**B**) roots after 12days of salt treatment. The scale (Log^2^ of the mean values after normalization; *n* = 3) shows the increase in relative concentrations from blue to red color, compared to control (0 h).

**Figure 10 ijms-23-01603-f010:**
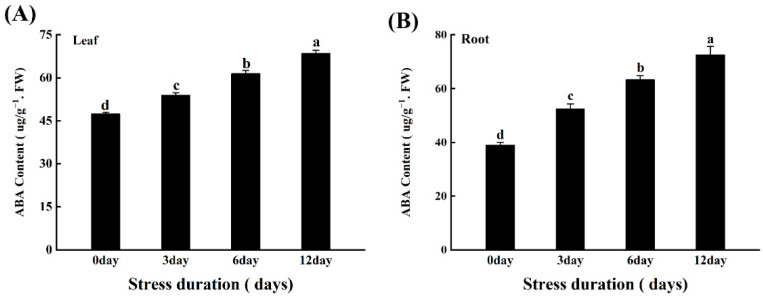
Effects of salt stress on Abscisic Acid in leaves (**A**) and roots (**B**) of tomato seedlings. Data was recorded after 12 days of growing tomato seedlings under control and salt stress for 12 days. Values are mean ± SD (Standard Deviation) of three replications. Different letters indicate significant differences at *p* ≤ 0.05 (Tukey’s HSD test).

**Figure 11 ijms-23-01603-f011:**
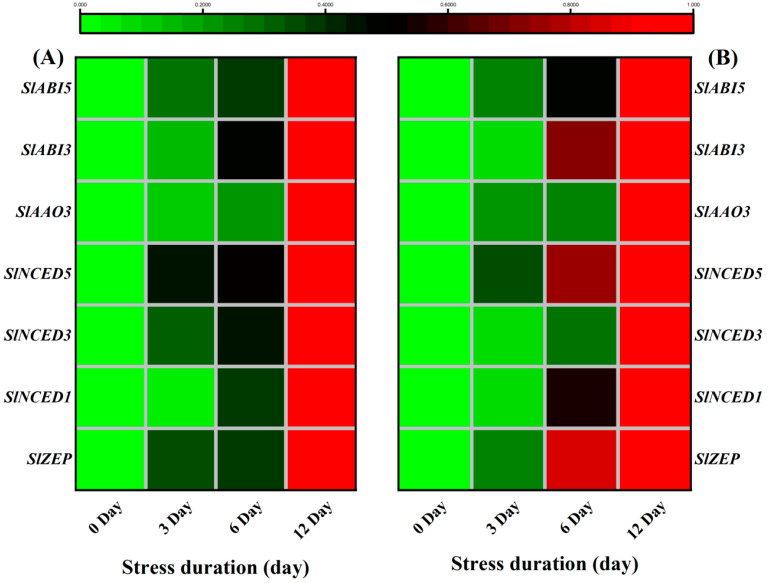
Heatmap of selected genes related to ABA biosynthesis pathway quantified from tomato (**A**) Leaves; (**B**) roots after 12 days of salt treatment. The scale (Log^2^ of the mean values after normalization; *n* = 3) shows the increase in relative concentrations from green to red color, compared to control (0 h).

**Figure 12 ijms-23-01603-f012:**
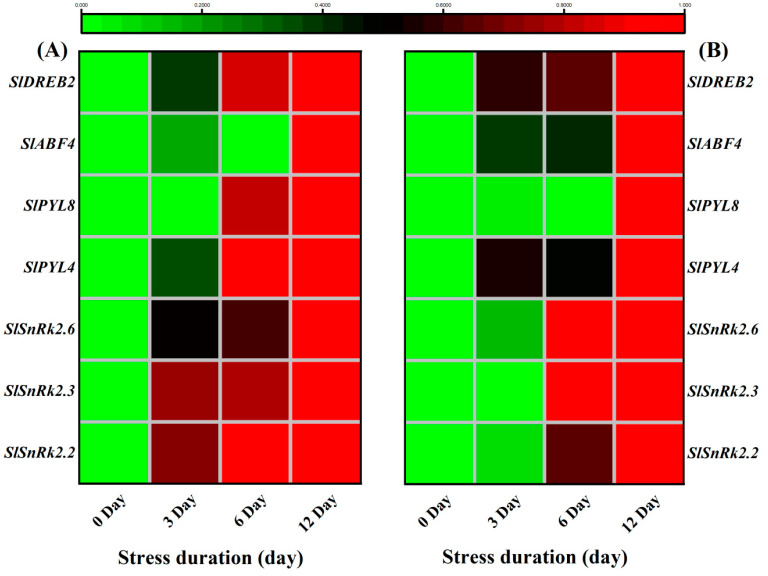
Heatmap of selected genes related to ABA signalling pathway. The relative expression level was quantified from tomato (**A**) Leaves and (**B**) roots after 12 days of salt treatment. The scale (Log^2^ of the mean values after normalization; *n* = 3) shows the increase in relative concentrations from green to red color, compared to control (0 h).

**Table 1 ijms-23-01603-t001:** Interactive effect of NaCl treatments on the macro and micronutrients uptake in leaves of tomato seedlings after 12 days under control and salanity stress.

Treatments	P	Zn	Fe	Mn	Mg	Ca^2+^	Cu	Na^+^	K^+^
0 day	6.62 ± 0.74 a	0.03 ± 0.00 b	1.46 ± 2.12 a	0.46 ± 0.08 a	3.16 ± 0.09 a	0.67 ± 0.03 b	4.61 ± 0.23 a	0.26 ± 0.01 d	22.78 ± 2.44 a
3 day	5.46 ± 0.08 b	0.04 ± 0.00 b	1.34 ± 5.75 ab	0.29 ± 0.01 b	2.58 ± 0.08 ab	0.84 ± 0.10 b	5.54 ± 0.69 ab	0.30 ± 0.01 c	18.66 ± 0.34 b
6 day	5.16 ± 0.28 b	0.04 ± 0.00 b	1.23 ± 8.09 b	0.19 ± 0.00 c	2.24 ± 0.07 b	3.78 ± 0.02 b	6.99 ± 0.66 ab	0.31 ± 0.01 b	17.64 ± 0.58 b
12 day	4.60 ± 0.08 b	0.05 ± 0.01 a	0.101 ± 12.90 c	0.08 ± 0.01 d	2.02 ± 0.02 b	5.96 ± 0.47 a	10.13 ± 4.00 b	0.35 ± 0.01 a	16.55 ± 0.55 b

Data represent the mean and standard deviation (SD) of three replications. Different letters indicate significant differences according to ‘Tukey’s HSD test at *p* ≤ 0.05.

**Table 2 ijms-23-01603-t002:** Interactive effect of NaCl treatments on the macro and micro nutrients uptake in roots of tomato seedlings after 12 day under control and salanity stress.

Treatments	P	Zn	Fe	Mn	Mg	Ca^2+^	Cu	Na^+^	K^+^
0 day	9.67 ± 0.27 a	0.01 ± 0.01 d	0.19 ± 0.01 a	1.15 ± 0.12 a	4.03 ± 0.39 a	1.79 ± 0.04 c	6.16 ± 0.52 c	0.25± 0.01 d	28.41 ± 1.28 a
3 day	8.68 ± 0.47 b	0.01 ± 0.01 c	0.15 ± 0.01 b	1.29 ± 0.03 b	3.65 ± 0.02 a	2.06 ± 0.17 c	6.79 ± 0.10 bc	0.30 ± 0.01 c	25.31 ± 0.16 b
6 day	5.53 ± 0.09 c	0.03 ± 0.01 b	0.12 ± 0.01 c	1.09 ± 0.01 c	2.87 ± 0.06 b	2.39 ± 0.06 b	7.35 ± 0.40 b	0.42 ± 0.01 b	22.21 ± 0.21 c
12 day	4.70 ± 0.34 d	0.04 ± 0.01 a	0.09 ± 0.01 d	0.90 ± 0.08 d	2.44 ± 0.17 b	3.08 ± 0.21 a	8.37 ± 0.33 a	0.47 ± 0.02 a	20.21 ± 0.89 d

Data represent the mean and standard deviation (SD) of three replications. Different letters indicate significant differences according to ‘Tukey’s HSD test at *p* ≤ 0.05.

## Data Availability

Excluded (because all the data and Appendix A have already been provided in the paper).

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
