# Peer review of "A Comprehensive Evaluation of Salt Tolerance in Tomato (Var. Ailsa Craig): Responses of Physiological and Transcriptional Changes in RBOH’s and ABA Biosynthesis and Signalling Genes"

_ijms, 2022, doi:10.3390/ijms23031603_

Round 1

Reviewer 1 Report

The present paper: ”Comprehensive evaluation of salt tolerance in tomato (Var. Ailsa Craig): Responses of Photosynthesis, Antioxidant defense system, ROS Production and transcriptional changes in RBOH’s and ABA biosynthesis and signalling genes” by Abdul Raziq, Yu Wang, Atta Mohi Ud Din, Jin Sun, Sheng Shu, Shirong Guo, describes the salinity response in tomato seedlings under progressive salt stress duration by , physiological, biochemical and transcriptomic analyses. The authors reported increased gene expression of antioxidant enzymes and ABA biosynthesis/signalling genes, suggesting a role of antioxidative system, RBOH genes and ABA to survive the salt-induced oxidative stress in tomato.

The results are interesting, they are clearly presented and well interpreted and commented in Discussion.

Some minor language mistakes should be revised.

In Fig 1, 2, 3, 4, 5 and 7 all the histograms should have the X axis title, in fig captions I suggest replacing “different alphabets” with “different letters”

In my opinion the paper is suitable for publication in International Journal of Molecular Sciences after minor revisions.

Author Response

All the suggested changes have been done according to the reviewer's comments.

Reviewer 2 Report

The authors describe the response of tomato seedlings under a salinity stress time course which last 12 days. It is interesting that the authors decided to extend the stress time course up to 12 days, considering that in most of the tomato salinity response experiments were lasted only for 48 or 72 hours. With regard to the central idea of the paper, this is a wide correlative study associating already know molecular responses such as antioxidant machinery, enzymatic activities of SOD, Catalase etc, ABA biosynthesis and signaling components and NADPH oxidase gene expression, lipid peroxidation, osmotic potential etc with  the salinity response. Although the authors associate their finding extensively with available literature and taking into consideration the fact that most of these response have already being investigated in tomato, it would be interesting to better associate the results to each other in order to be presented that this response at the molecular level is a coordinated effort to keep cell homeostasis and survive long term salinity stress. Otherwise the whole experimental effort seems to be fragmented in different subcategories. 

Author Response

(The authors gave the same response as above.)

Author Response

(The authors gave the same response as above.)
